Mapping the self-association domains of ataxin-1: identification of novel non overlapping motifs

Menon Rajesh P. 1 rmenon@nimr.mrc.ac.uk
Soong Daniel 2 3
de Chiara Cesira 1
Holt Mark 2
McCormick John E. 1
Anilkumar Narayana 3
Pastore Annalisa 1 4 apastor@nimr.mrc.ac.uk
1 MRC National Institute for Medical Research, The Ridgeway , London , UK
2 Randall Division for Cell and Molecular Biophysics, New Hunt’s House, King’s College London , Guy’s Campus, London , UK
3 British Heart Foundation Centre of Research Excellence, King’s College London , Denmark Hill Campus, London , UK
4 Department of Molecular Neuroscience, Institute of Psychiatry, King’s College London , Denmark Hill Campus, London , UK
Kostyukova Alla
Electronic publication date: 2014 Mar 25
Publication date: 2014
Volume: 2
Electronic Location ID: e323
Received 2014 Jan 29; Accepted 2014 Mar 7
Copyright: © 2014 Pastore et al.
Copyright year: 2014
Copyright holder: Pastore et al.
License: This is an open access article distributed under the terms of the Creative Commons Attribution License, which permits unrestricted use, distribution, reproduction and adaptation in any medium and for any purpose provided that it is properly attributed. For attribution, the original author(s), title, publication source (PeerJ) and either DOI or URL of the article must be cited.
License URL: https://creativecommons.org/licenses/by/4.0/

Keywords: Foci, Misfolding diseases, FRET, Confocal microscopy, Spinocerebellar ataxia type 1

Funding: MRC ref. U117584256 The work was supported by the MRC (Grant ref. U117584256), from Ataxia UK and from the British Heart Foundation. The funders had no role in study design, data collection and analysis, decision to publish, or preparation of the manuscript.

==============================
The neurodegenerative disease spinocerebellar ataxia type 1 (SCA1) is caused by aggregation and misfolding of the ataxin-1 protein. While the pathology correlates with mutations that lead to expansion of a polyglutamine tract in the protein, other regions contribute to the aggregation process as also non-expanded ataxin-1 is intrinsically aggregation-prone and forms nuclear foci in cell. Here, we have used a combined approach based on FRET analysis, confocal microscopy and in vitro techniques to map aggregation-prone regions other than polyglutamine and to establish the importance of dimerization in self-association/foci formation. Identification of aggregation-prone regions other than polyglutamine could greatly help the development of SCA1 treatment more specific than that based on targeting the low complexity polyglutamine region.

Introduction

The inherited disease spinocerebellar ataxia type 1 (SCA1) is an autosomal dominant neurodegenerative pathology characterized by progressive loss of Purkinje cells in the cerebellar cortex and of neurons in the spinocerebellum (Zoghbi & Orr, 1995; Cummings, Orr & Zoghbi, 1999; Matilla-Dueñas, Goold & Giunti, 2008). The pathogenic mechanism of SCA1, presently incurable, seems to be complex (de Chiara & Pastore, in press). It is thought to be caused by aggregation and misfolding of ataxin-1 that is associated to expansion of a polymorphic polyglutamine (polyQ) tract in the N-terminus of the protein (Orr et al., 1993; Cummings et al., 1998; Klement et al., 1998; de Chiara et al., 2005; Mizutani et al., 2005; Tsuda et al., 2005; Lam et al., 2006). Recent results suggest that damage to the nuclear membrane caused by the pathogenic ataxin-1 could eventually lead to cell death (Mapelli et al., 2012). A similar mechanism of polyQ expansion triggers the aggregation of a larger family of polyQ containing proteins such as the better known Huntington’s chorea (Arrasate & Finkbeiner, 2012). For all members of this disease family, polyQ expansion seems to be the necessary event for disease development (Verbeek & van de Warrenburg, 2011; Robertson & Bottomley, 2012; Blum, Schwendeman & Shaham, 2013; Menon et al., 2013). It is, however, an accepted view that regions outside the polyQ tracts significantly contribute to the aggregation process urging the importance of studying protein context and analysing the behaviour of regions of these proteins that are also sequence-wise distant from the polyQ tract (Masino et al., 2004; de Chiara et al., 2005; Ellisdon, Thomas & Bottomley, 2006; Gales et al., 2005).

In agreement with this view, ataxin-1 is an intrinsically aggregation-prone protein known to form, also in its non-expanded form, diffuse cellular aggregates sometimes named foci (Matilla et al., 1997; Tsai et al., 2004; de Chiara et al., 2005; Osmand, Berthelier & Wetzel, 2006; Menon et al., 2012). The size of the foci increases in the presence of polyQ expansion. A self-association region of the non-expanded protein was mapped in the centre of the protein and identified to overlap with the globular AXH domain that spans residues 562–689 (SMART SM00536) (Burright et al., 1997; de Chiara et al., 2003) of the otherwise mostly unstructured ataxin-1 protein (de Chiara & Pastore, 2011). This motif is functionally very important as it is involved in transcriptional regulation as well as in the RNA-binding activity of ataxin-1 (Matilla et al., 1997; Okazawa et al., 2002; Tsai et al., 2004; de Chiara et al., 2003; de Chiara et al., 2005; Mizutani et al., 2005; Tsuda et al., 2005; Lam et al., 2006; Serra et al., 2006; Goold et al., 2007; Lee et al., 2011). AXH is also involved in the majority of the known interactions of ataxin-1 with other proteins, most of which are transcriptional regulators (Tsai et al., 2004; Tsuda et al., 2005; Lam et al., 2006; Goold et al., 2007; Serra et al., 2006). Although AXH does not contain a polyQ tract and is sequence-wise distant from it, it seems to play an important role in ataxin-1 aggregation. In solution, the isolated AXH forms a complex equilibrium between monomer, dimer, tetramer and higher molecular weight species (de Chiara et al., 2013a). This process was suggested to be on-pathway to protein aggregation and fibre formation. In further support to this theory, deletion of the AXH domain leads to reduction of intra-nuclear aggregate formation by expanded ataxin-1 in eukaryotic cells (de Chiara et al., 2005).

Realization that ataxin-1 aggregation may be triggered by more than one region has suggested that this behaviour could inspire the development of new drugs that could target regions other than or in addition to the polyQ tract. Such drugs would be potentially more specific than compounds preventing the aggregation of the low complexity polyQ. A recent report has, for instance, shown that stabilization of the monomeric form of the AXH domain by formation of a complex with a peptide from the natural partner protein Capicua (CIC) prevents the aggregation and misfolding of the isolated domain (de Chiara et al., 2013a; de Chiara et al., 2013b). If the same held true also for the full-length protein, this strategy would have terrific consequences for the design of novel therapeutic lead compounds. To follow up this strategy, however, more information about the regions responsible for self-association is needed.

Here, we have combined studies in cell using confocal microscopy as well as FRET (Förster Resonance Energy Transfer) analysis and in vitro investigations of isolated regions of ataxin-1 to map the regions necessary for foci formation and explore the relationship between dimerization and self-association of non-expanded ataxin-1. FRET based approaches have proven to be a powerful tool for the analysis of protein homo-dimerization in cells (Itoh et al., 2011; Placone & Hristova, 2012; Hlavackova et al., 2012). Our results establish the existence of dual non-overlapping self-association motifs within ataxin-1 reinforcing the importance of the AXH domain in self-assembly. We also demonstrate that destabilization of AXH dimerization appreciably reduces protein self-association. This evidence may pave the way to new directions towards the development of anti-SCA1 drug design.

Materials and Methods

Plasmids, cell culture, transfections and imaging

Non-expanded (Q30) ataxin-1 and truncated ataxin-1 fusion proteins were constructed in pEYFP or pECFP vectors using the standard PCR and mutagenesis methods previously established in our laboratory (Menon et al., 2012). COS cells were grown in chamber slides in Dulbecco’s modified Eagle medium supplemented with 10% foetal bovine serum and 100 U/ml penicillin-streptomycin (Invitrogen Life Technologies). Cells were transfected with appropriate plasmid DNA using GeneCellin tranfection reagent (BioCellChallenge). Cells were fixed using 4% paraformaldehyde 54 h post-transfection and slides were mounted using CitiFluor (Agar Scientific). Cells were observed and recorded using a laser scanning confocal microscope (de Chiara et al., 2009).

Analytical size exclusion chromatography

Size exclusion chromatography was performed using a prepacked Superdex-75™ 10/300 GL column (Pharmacia) equilibrated with a 20 mM Tris–HCl pH 7, 150 mM NaCl, 1 mM TCEP buffer solution. Aliquots of 200 µl of 150 µM AXH and TLND2AXH incubated for 24 h at 37 °C in 20 mM pH 7, 150 mM NaCl, 1 mM TCEP were injected separately and eluted using a 0.8 ml/min flow rate. Albumin (67 kDa), Ovalbumin (43 kDa), Carbonic anhydrase (29 kDa), and Ribonuclease A (13.7 kDa) were used as standards for the molecular mass, whereas the Blue Dextran 2000 was used for the determination of the void volume of the column.

FRET microscopy

Samples for FRET were imaged on a Zeiss LSM 510 confocal microscope using a 63× 1.4NA Plan NeoFluar oil immersion objective and FRET analysis was carried out as previously described in detail (Menon et al., 2012). Pre- and post-bleach CFP and YFP images were imported into Mathematica 7.0 for processing as described (Matthews et al., 2008). Briefly, images were smoothed using a 3 × 3 box mean filter, background subtracted, and post-bleach images fade compensated. E=CFPpostbleach−CFPprebleachCFPpostbleach

FRET efficiencies were then extracted from pixels falling inside the bleach region and plotted against the bleach efficiency on a pixel-by-pixel basis. FRET efficiency showed a linear correlation with bleach efficiency enabling determination of FRET efficiency at 100% bleach efficiency by extrapolation. Data from images were used only if YFP bleaching efficiency was greater than 50%. Finally, the FRET efficiency was converted in to the inter-fluorophore radius using: r=Ro1E−16

where Ro is the Förster radius for CFP and YFP, which is 4.95 nm.

Results

Ataxin-1 foci formation is independent of polyQ and mediated by the C-terminus

We first explored the relationship between foci formation and polyQ in non-expanded ataxin-1. We used non-expanded ataxin-1 to be able to detect the intrinsic properties of the functional protein that could then be transferred to the expanded form. We created several deletion constructs for expression in mammalian cells in which the protein was N-terminally attached to the yellow fluorescent protein (YFP) (Fig. 1). The first of these mutants, hereafter termed NT, contains the N-terminus of ataxin-1 up to the start of the polyQ tract. Its behaviour was compared with that of a mutant (termed Atx1ΔNT) that excludes the region preceding the polyQ (30Qs) tract. The results from the whole analysis are summarized in Table 1.

Figure 1 Summary of the ataxin-1 constructs used in the present study.

Full-length ataxin-1 protein is represented by a black line. The positions of the polyQ tract, the AXH domain and NLS are explicitly indicated.

Table 1 Ataxin-1 constructs and their foci forming abilities.

Construct	Amino acids involved	Foci formation (Yes or No) and
comparison of foci size
with ataxin-1 wild-type	Foci localization	
Ataxin-1 wild-type	1–816	Yes and N/A	Nucleus	
NT	1–196	No	N/A	
NTQ	1–226	No	N/A	
Atx1ΔNT	197–816	Yes and similar	Nucleus	
Atx1ΔNTQ	227–816	Yes and similar	Nucleus	
TLND2End	410–816	Yes and similar	Nucleus	
AXH2End	562–816	No	N/A	
AXH	562–689	No	N/A	
CT2AXH	227–561	Yes and similar	Cytoplasm	
TLND2AXH	410–561	Yes and smaller	Cytoplasm	
Atx1ΔCT2AXH	Deletion of 227–561	No	N/A	

As expected, wild-type ataxin-1 fused to YFP readily formed nuclear foci (Figs. 2A and 2B). The YFP fusions of ataxin-1 constructs were similar in expression pattern to equivalent constructs that lacked YFP fusion and were stained with antibodies (data not shown), suggesting that the presence of YFP does not significantly influence the data. The NT construct showed a diffused pattern of expression, demonstrating that this region is not involved in foci formation (Fig. 2C). Addition of the polyQ tract to the N-terminal region (construct named NTQ) did not alter the behaviour of the cells (Fig. 2D). We observed that constructs Atx1ΔNTQ and Atx1ΔNT were both able to form foci (Figs. 2E and 2F).

Figure 2 Identifying the region responsible for foci formation in ataxin-1.

Various deletion constructs of ataxin-1 were expressed as YFP fusion proteins in COS cells. (A) Full-length ataxin-1 30Q-YFP. Overlay with DAPI is shown in B. Expression analysis shows a diffused expression pattern in cells expressing ataxin-1 N terminal regions without (C) or with the polyQ tract (D). In contrast, cells expressing the C-terminal regions either without (E) or with (F) the polyQ tract readily formed foci.

The observation that Atx1ΔNTQ, which lacks the polyQ tract, is able to form foci demonstrates that polyQ region is not per se a major foci forming factor. These results thus indicate that the C-terminus of ataxin-1 alone is involved in foci formation which does not involve the polyQ tract.

The AXH domain is insufficient to form foci but needs N-terminal extension

In order to further map the region required for nuclear foci formation, we expressed deletion mutants from the C-terminal region of ataxin-1 fused with YFP. We first expressed the region starting at the AXH domain and ending at the last residue of ataxin-1 (AXH2End, residues 562–816), which also contains the endogenous Nuclear Localisation Signal (NLS). As expected, this protein was mostly nuclear, but expressed itself in a diffused pattern (Figs. 3A and 3B). A diffused pattern of nucleo-cytoplasmic expression was also observed for a construct with the AXH domain alone starting at amino acids ASPAA and comprising residues 562–689, termed AXH (Figs. 1, 3C and 3D). Foci formation was observed instead when the AXH2End construct was N-terminally extended (residues 410–816, termed TLND2End, where TLND represents the N-terminal amino acid sequence of the construct) (Figs. 3E and 3F).

Figure 3 AXH domain is not involved in foci formation.

A YFP construct starting at the ataxin-1 AXH domain and ending at the last ataxin-1 amino acid (AXH2End) was expressed as a YFP fusion in COS cells (A and B). These cells showed a diffused YFP fluorescence which was nuclear, as evidenced by DAPI overlay (B). YFP tagged AXH domain also failed to form foci (C and D). Upon N-terminal extension of the AXH2End construct to include further residues starting from amino acids TLND, nuclear foci formation was observed (Figs. 4E and 4F). YFP fluorescence in left panels is overlaid with DAPI in right panels.

These results thus suggest that the aggregation-prone AXH domain is insufficient for foci formation by its own.

Evidence for the self-association prone CT2AXH motif

Since TLND2End formed foci while AXH2End did not, we next expressed the TLND2AXH region (residues 410–561) as YFP fusion to analyse if this is independently capable of foci formation. We observed that this construct formed foci that were, however, smaller in size as well as in number (Figs. 4A–4C). This construct appeared to express mainly in a diffused form, but compared to other diffusedly expressing constructs such as the AXH, we could clearly observe small foci. Therefore this construct appeared to have a behaviour intermediate between those forming foci and those that do not. N-terminal extension of TLND2AXH (i.e., residues 227–561, construct CT2AXH) showed that the CT2AXH construct enhances foci formation ability (Figs. 4D and 4E). The foci in both instances were mostly extranuclear, which is not surprising as this region is not known to possess a functional NLS. C-terminal addition of SV40 NLS to CT2AXH resulted in exclusively nuclear foci formation (Figs. 4F and 4G). Lastly, deletion of the CT2AXH region from full-length YFP ataxin-1 (construct Atx1ΔCT2AXH) abolished foci formation (Figs. 4H and 4I).

Figure 4 Foci forming analysis of the CT2AXH region.

Residues TLND to AXH from ataxin-1 in fusion with YFP forms small foci that are extranuclear (A and B). One cell from the image is enlarged in C for clarity. N terminal extension of this region where the construct started after the polyQ tract (CT2AXH) formed larger foci which also were extranuclear (D and E). Addition of an NLS to this construct resulted in nuclear foci formation (F and G). Deletion of CT2AXH from ataxin-1 resulted in diffused expression of the protein (H and I). A, D, F and H show YFP fluorescence. YFP fluorescence is overlaid with DAPI in B, C, E, G and I.

These results show that foci formation takes place independently from the AXH domain and that the nuclear localization signal has an influence on foci localisation.

Identification of a self-association motif outside the polyQ region and the AXH domain

The isolated AXH domain is known to dimerize in vitro (de Chiara et al., 2003; Chen et al., 2004; de Chiara et al., 2005; de Chiara et al., 2013a). Dimerization rather than other forms of self-association could then be the seeding event for foci formation. Since nothing is known about the region N-terminally upstream to the AXH domain, we further characterized the TLND2AXH motif (residues 410–561) by analytical size exclusion chromatography (SEC) techniques to better understand the relationship between foci formation and the properties of the individual domains (Fig. 5). Our original intention was to do the experiment with the CT2AXH construct. However, we ran into problems of expression as the protein went into inclusion bodies. For this reason we switched to TLND2AXH, which expressed well and in soluble form.

Analytical gel-filtration chromatograms indicate that, in analogy with the AXH domain, the construct TLND2AXH is in equilibrium between two species which have an elution volume of 11.0 and 12.0 ml, corresponding to roughly 36 kDa and 30 kDa respectively if it were a globular protein (from the standards). Lower retention volumes are observed for TLND2AXH as compared to AXH despite their similar molecular weight. This would be justified by the elongated shape of the putatively unfolded TLND2AXH construct (de Chiara & Pastore, 2011) as compared to the compact fold of the AXH domain (Chen et al., 2004).

These results provide solid evidence of the existence of a newly identified self-association region within ataxin-1 that is independent from the AXH domain.

The two distinct dimerization motifs are independently capable of self-association in cells

To complement these in vitro studies of dimerization, we next tested the self-association regions identified in cultured cells and explored if they are capable of direct interaction leading to self-association also in cell. To verify that the full-length ataxin-1 protein is indeed capable of self-association, we expressed full-length non-expanded ataxin-1 as CFP and YFP fusions and carried out FRET analysis. Self-association of ataxin-1 was evident from positive FRET signal (Figs. 6A–6F). We observed comparable FRET signals both when full-length ataxin-1 was expressed as C-terminal fusion to CFP (CFP-C-Atx1, Fig. 6A) and YFP (YFP-C-Atx1, Fig. 6B) and when ataxin-1 was C-terminal to CFP (CFP-C-Atx1, Fig. 6D) and N-terminal to YFP (YFP-N-Atx1, Fig. 6E). This suggests that the position of the fluorophore in ataxin-1 does not influence the FRET signal significantly.

Figure 5 Analytical size exclusion chromatography.

Analytical gel-filtration chromatograms of AXH (continuous line) and TLND2AXH (dotted line) constructs. The position of molecular weight markers is indicated for comparison. Calculated molecular weights are 13.9 kDa for AXH and 16 kDa for TLND2AXH.

Figure 6 ‘Rainbow’ pseudocolour look-up table (LUT)-encoded pre- and post-bleach images of CFP and YFP fusion proteins.

Magnified crops of both CFP and YFP signals in the bleach region (black circles) are depicted for pre- and post-bleach for each FRET pair (C, F, I, L, O and R). All scale bars are 5 µm. The FRET pairs are, (A–C) CFP-C-Atx1 vs YFP-C-Atx1; (A) CFP fluorescence, (B) YFP fluorescence; (D–F) CFP-C-Atx1 vs YFP-N-Atx1; (D) CFP fluorescence, (E) YFP fluorescence; (G–I) CFP-567-Atx1 vs YFP-wt-Atx1; (G) CFP fluorescence, (H) YFP fluorescence; (J–L) CFP-CT2AXH vs YFP-CT2AXH; (J) CFP fluorescence, (K) YFP fluorescence; (M–O) CFP-AXH vs YFP-AXH; (M) CFP fluorescence, (N) YFP fluorescence; (P–R) CFP-TLND2AXH vs YFP-TLND2AXH; (P) CFP fluorescence, (Q) YFP fluorescence.

We then expressed the newly identified dimerization motif TLND2AXH as CFP and YFP fusions. Self-association was evident in FRET experiments using the TLND2AXH construct (Figs. 6P–6R). As expected, the N-terminal extended form of TLND2AXH (CT2AXH) was also found to interact directly in cells (Figs. 6J–6L). Similarly, we carried out FRET analysis with CFP and YFP fusions of the AXH domain. The results confirmed direct interaction between the YFP and CFP fusion proteins (Figs. 6M–6O).

Testing the effects of destabilization of AXH dimerization in cell

Finally, we used the FRET based approach to test the hypothesis that stabilization of the AXH domain into its monomeric form could result in reduction of protein aggregation. We used FRET analysis on co-expressed ataxin-1 YFP vs ataxin-1 CFP where ataxin-1 YFP plasmid also expressed a Capicua (CIC) peptide (residues 34–48) of the protein that we have recently been shown to interact with the AXH domain with high affinity and to force it in a monomeric form (de Chiara et al., 2013b). Unfortunately, we were unable to verify the effect of the peptide since the antibodies raised against it did not recognize the peptide (data not shown) preventing confirmation of peptide expression.

We resorted to a different strategy as a proof of principle. We have recently reported that mutation of a glycine at position 567 keeps the AXH domain in a predominantly monomeric form (de Chiara et al., 2013a). We thus tested if self-association in cells is affected by this mutation. We reasoned that a direct comparison is possible as this is only a point mutation which is unlikely to influence FRET data. We carried out a FRET analysis using the CFP ataxin-1 567 mutant and YFP-wild-type ataxin-1. Interestingly, we observed a significant reduction in the FRET signal (CFP 567 Atx1 versus YFP wt Atx1, Figs. 6G–6I and 7). This result suggests that identifying and manipulating vulnerable regions of ataxin-1 could be a tool to reduce self-association and possibly also have an effect on ataxin-1 aggregation. The corrected FRET efficiencies obtained with the different protein pairs and combined from different photo-bleaching experiments were calculated and are summarised in Fig. 7. We found that the FRET signal of CT2AXH was much higher as compared to full-length ataxin-1. This may be due to the fact that the CT2AXH fragments are smaller than full-length ataxin-1, the fluorophores are likely to be closer to each other and the radial distance of the resonance energy transfer is accordingly smaller, leading to stronger FRET. We cannot also rule out the possibility that the truncated fragments tend to bind more strongly to each other. Reduced FRET of TLND2AXH, compared to CT2AXH is expected as the smaller TLND2AH is obviously forming smaller foci—pointing towards the possibility of a weaker association.

Figure 7 Box and whisker plots depicting population distribution of percentage corrected FRET and showing maximum, minimum, upper and lower quartiles, and sample median.

The individual FRET pairs are shown in the X axis. These are: (1) CFP-C–Atx1 vs YFP-C-Atx1; (2) CFP-C-Atx1 vs YFP-N-Atx1; (3) CFP-567-Atx1 vs YFP-wt-Atx1; (4) CFP-CT2AXH vs YFP-CT2AXH; (5) CFP-AXH vs YFP-AXH; (6) CFP-TLND2AXH vs YFP-TLND2AXH. Means ± standard errors, rounded to one decimal place, are shown above each boxplot. Statistical significance bars are shown and represent results of unpaired t-tests of mean difference = 0 and represent number of individual bleach events pooled from at least 4 individual cells.

We can thus conclude that both dimerization domains of ataxin-1 contribute to the self-interaction of the protein in cells.

Discussion

One aspect that distinguishes ataxin-1 from most other proteins is its ability to form small dense nuclear bodies (Skinner et al., 1997; Matilla et al., 1997). These bodies have been variously described as nuclear structures (Skinner et al., 1997), foci (Tsai et al., 2004), inclusions (Dovey et al., 2004), nuclear accumulations (Krol et al., 2008), or aggresomes (Latonen, 2011) with little consensus on the terminology used. Here, we preferred to name these structures foci as they are formed not just by the mutant expanded protein but also by the non-expanded protein or even by ataxin-1 without the incriminating polyQ tract (Tsai et al., 2004). This is in agreement with the observations of nuclear structures both for expanded and non-expanded ataxin-1 which only differed in size (Skinner et al., 1997). Regardless of the terminology, there is little doubt that these ataxin-1 structures are important in the normal as well as in pathological aspects of ataxin-1 function as various functional partners have been shown to associate with ataxin-1 within these bodies (Matilla et al., 1997; Chen et al., 2003; Mizutani et al., 2005; Menon et al., 2012). The small nuclear foci have been shown to further merge into larger bodies and this phenomenon is accelerated by polyQ expansion (Krol et al., 2008). Therefore the foci might also serve as seeding ground for aggregate formation. These observations suggest that new insights into foci formation events and identification of the regions responsible for this phenomenon could help not only the elucidation of the normal function of ataxin-1, but also the development of therapeutic strategies.

Our in situ investigations have now revealed that non-expanded ataxin-1 has two self-association motifs. Both AXH domain and the TLND2AXH motif are able to self-associate in transfected cells, as shown by FRET analysis. An earlier description of ataxin-1 self-association had identified this property almost exclusively with the AXH domain (Burright et al., 1997). Accordingly, subsequent investigators have described the self association region as partially overlapping the AXH domain (see for instance Krol et al., 2008). We have now shown that this is only partially the case. FRET analysis, compared to co-localisation analysis, is capable of better demonstrating direct protein–protein interaction and is highly specific for self-association independently from the presence of other macromolecules. We have shown that both the non-overlapping TLND2AXH and AXH domains are independently capable of self-association while the former but not the latter is essential for foci formation. These observations are further supported by in vitro analysis of the dimerization properties of these motifs. Interestingly, gel filtration analysis demonstrates that the TLND2AXH motif is a dimer in equilibrium with the monomer, much like the AXH domain itself (de Chiara et al., 2013a). Collectively, these results reiterate the importance of non-polyQ elements in the ataxin-1 functions and suggest that the dimerization observed for the AXH domain is an on-pathway event to aggregation.

In the attempt to define the relationship between dimerization and self-association, we also questioned if events that potentially reduce dimerization of individual motifs are able to influence the self-association tendency of ataxin-1 as measured by FRET analysis. In light of a recent report (Kim et al., 2013) and of our own findings (de Chiara et al., 2013b), we reasoned that a CIC peptide similar to the one used in our studies (de Chiara et al., 2013b) might reduce the self-association of ataxin-1. We therefore tried to do FRET analysis of CFP ataxin-1 in the presence of the CIC peptide (or a control scrambled peptide) expressed from a pIRES vector that also expressed YFP ataxin-1. However, we were unable to confirm the expression of the peptides as the antibodies raised against them were not effective. As an alternative to test our hypothesis, we carried out a FRET analysis using an ataxin-1 single mutant (A567G). A similar mutation in the AXH domain has been shown to keep the AXH domain in a predominantly monomeric form in vitro (de Chiara et al., 2013a). The FRET analysis between ataxin-1 567 mutant and ataxin-1 wild-type protein indicated a significantly lower FRET signal compared to wild-type/wild-type ataxin-1 FRET. Therefore, the AXH domain does not seem to contribute significantly in foci formation but has a role in self-association in agreement with previous data which showed that deletion in full-length ataxin-1 of the domain reduces intracellular aggregation (de Chiara et al., 2005). Our results provide a proof of principle of the potential effectiveness that can be achieved by disrupting ataxin-1 dimerization.

Conclusions

In conclusion, our study contributes to clarify the self-association properties of ataxin-1 and the relationship between these and the dimerization observed at the level of individual domains. This information may be used in further studies to probe the effect of the CIC interactions on aggregation and be helpful for designing new approaches to SCA1 therapy.

Additional Information and Declarations

Competing Interests

Author Contributions

The authors declare there are no competing interests. Annalisa Pastore is an Academic Editor for PeerJ.

Rajesh P. Menon and Cesira de Chiara conceived and designed the experiments, performed the experiments, analyzed the data, contributed reagents/materials/analysis tools, wrote the paper, prepared figures and/or tables.

Daniel Soong conceived and designed the experiments, performed the experiments, analyzed the data, contributed reagents/materials/analysis tools, prepared figures and/or tables, reviewed drafts of the paper.

Mark Holt analyzed the data, contributed reagents/materials/analysis tools, reviewed drafts of the paper.

John E. McCormick performed the experiments, contributed reagents/materials/analysis tools, reviewed drafts of the paper.

Narayana Anilkumar performed the experiments, analyzed the data, contributed reagents/materials/analysis tools, reviewed drafts of the paper.

Annalisa Pastore analyzed the data, wrote the paper.

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
