# Peer review of "Mapping the self-association domains of ataxin-1: identification of novel non overlapping motifs"

_PeerJ, doi:10.7717/peerj.323_

## Round 0.1 · original submission · Minor Revisions

Please, address the reviewer's comments.

·

Basic reporting

No Comments

Experimental design

No comments

Validity of the findings

This work is very interesting and is presented with well-constructed and rigorous experiments of high technical standard. It shed light on a new functional domain of ataxin-1.
There is a purely speculative aspect I wold like to discuss. The characterization of two functional domains, AXH and TLND2DAXH, is extremely interesting for the understanding of the SCA1 gene function and its possible relationship/interaction with other proteins. This allows us to understand how the aggregation function can influence the clinical picture caused by the presence of expanded polyglutamine tracts.
The disease, however, is not a direct function of the aggregation, which is rather considered a defence from toxic molecules by several researchers. In fact, transgenic mice containing a deletion within the “self-association region” (partly overlapping with those presented here) developed ataxia and Purkinje cell pathology similar to the original SCA1 mice. No evidence of nuclear ataxin-1 aggregates was found in the Purkinje cells of these mice (Klement et al, Cell, 95, 41–53, 1998). Moreover, while the aggregates have been also found in cells carrying normal alleles, cell death only occurred in the presence of an expanded polyglutamine tract, as demonstrated by several in vitro and in vivo experiments. Therefore, it is likely that the final and more severe step of SCA1, and more in general polyglutamine diseases, rely on a toxic function gained by the expanded polyglutamine protein portion. According to this hypothesis, it has been demonstrated that an interaction between pathological molecules (expanded polyQ peptide or full length ataxin-1) and the nuclear membrane is responsible for this end point of cell degeneration (Mapelli et al, Biochimica et Biophysica Acta 1822 (2012) 906–917).
Therefore, the interesting findings presented in the Pastore and co-authors’ paper could help for the development of therapeutic strategies, but they must be designed taking into account that SCA1 neuropathology is a complex disease caused by multiple factors.

·

Basic reporting

This work reports on a study of the self-association of a non-pathogenic form of the protein Ataxin-1. Overall the reporting is good with a logical structure. However, the presentation is not the most effective and there are room for improvement. The article may benefit from more vigorous English grammar checking. There are sentences that are cumbersome and straining and needs rewriting. Some figures, labels, and symbols used can be improved to enhance clarity.

Experimental design

The methods used are appropriate and designed to give answer to the biological question under investigation. For in vitro protein interaction studies, fluorescence-labelled fusions were used to monitor protein localisation and foci formation inside the cell. This was used to identify the self association regions for foci formation. Protein fragments were checked for interactions in vitro and in vivo with the FRET method which is a sensitive and well established method.

Validity of the findings

The interpretation of experiments are in general solid and sound and the conclusions are well justified.

[1] I have some concerns about the way the authors using the terms “sufficient” and “necessary” rather loosely.

p.4, line 6 “AXH is also necessary and sufficient for the majority …”
Most interaction studies (Y2H, GST pulldown) define what is “necessary” but cannot conclude if one component is “sufficient”.

p.5, line 3 “necessary/sufficient”
What does the stroke mean?

“The AXH domain is insufficient to form foci …” (Title of Section)
I agree with this. The AXH needs to be extended N-terminally to form foci, according to this section’s results. But the last sentence, “… the aggregation-prone AXH domain does not have a predominant role…” is not supported by these results. The results in this section suggest that AXH is not sufficient, but nothing suggest that it is unimportant.


[2] In the discussion of TLND2AXH, it was said to “formed foci that were, however, smaller in size as well as in number.” I agree with this. However, the authors should also discuss that a lot of the fluorescence remains diffused like those formed by other constructs that do not form foci at all. (Fig. 4A,B,C) I think this is actually a very interesting construct because it captures something in between, able to form foci but not all of these do. So, while all other constructs are all-or-none (foci or diffuse), this is a ~half-and-half. And the rest of the article should discuss referring to this result as such, including the respective entry in Table 1.

[3] P.9 Section Ataxin-1 contains dual dimerization motifs
I don’t think this section title summarise well the content here.

[4] While CT2AXH has no doubt more convincingly displayed foci-forming; TLND2AXH shows only partial foci-forming and small foci, suggesting that the interaction may be incomplete. So TLND2AXH is less representative of the protein-protein self interaction of Ataxin-1. I do not understand why the authors chose to work with TLND2AXH rather than CT2AXH for SEC.
SEC results: This is the paragraph which I found most problematic. It is well documented that unstructured proteins run at different elution volumes to those calibrated by globular proteins. Without detailing how to calibrate (e.g. there were previous works on how to correct for elution volumes for unfolded proteins) the TLND2AXH (assuming it is unfolded), one cannot really analyse the results and conclude that it “is in equilibrium between two species… “ and “has an elution volume compatible with that of a monomer and a dimer.” Also the retention volume is lower not higher for TLND2AXH. It does not make sense to compare the elution volumes of a folded and an unfolded protein and account for the difference in calculated molecular masses.

“The elution volume (12.7 ml) of the higher molecular weight species of TLND2AXH…”
From what I can read, the 2 peaks of TLND2AXH are 11 ml and 12 ml. The 12.7 ml peak belongs to AXH.

“…significantly smaller that expected for a trimer or a tetramer.” As explained above, need more vigorous calibration to discuss the estimated molecular weight of an unfolded protein.

From these discrepancies, it seems likely that the authors have mislabelled the 2 samples in Fig.5 and have the 2 proteins swapped. Everything makes a bit more sense if the continuous line represents TLND2AXH, except that the molecular weight estimation of the unfolded TLND2AXH still demands more vigorous treatment rather than descriptively as it is.

p.12, last 2 lines. Discussion about gel filtration results here should be amended to reflect whatever the authors have revised according to the comments above.


[5] The authors did not give any evidence to support that TLND2AXH is unfolded.

[6] p.12, 2nd paragraph, line 5 “This incorrect assumption has lead …”
I think it is quite harsh to say that an earlier work aligning self association to exclusively within the AXH domain was an“incorrect assumption”. After all, supported by this work, AXH domain does contribute significantly, although not exclusively. I suggest the authors to reword here. And then in line 7 “…clearly shown that this is not the case”. For the same reasons, it is not entirely “not the case” but is “half a case”. I suggest to reword here too.


[7] dimerization and self-association

p.10 line 3 “dimerization/self-association” What does the stroke mean?

To my understanding, dimerization and self-association is the same thing. While there is no attempt in this study to further characterise the order of the oligomerisation state, they mean the same. In some places, these two terms were used like they are different states. E.g. p.13 lines 4-6. What exactly is the difference between these two terms? I have an impression that the authors describe the interaction observed in the crystal structure of the AXH domain as dimerization and everything else self-association. However, dimerization is just one specific form of self-association. Again, in the conclusion this differentiation was made again. Can the authors clarify this a bit clearer --- defining what they mean by self association in the text as in its difference to dimerization.

[8] More discussion of Fig.7 is needed. Why does CT2AXH lead to increased FRET? It is worth mentioning that both AXH and TLND2AXH individually are slightly reduced compared to full-length Atx1.

Additional comments

The following are specific comments in the text, mainly concerning bad use of English and figures.

p.2, line 1, “The neurodegenerative spinocerebellar …”
Missing the word “disease”?

p.3, first paragraph, last but 5 line “It is however an accepted concept that…”
May be use “view” instead of “concept”?

p.3, 2nd paragraph, line 2 “…diffuse cellular aggregates some time named”
“sometimes” not “some time”

p.3, last line “… to overlap the only certified globular…”
“overlap with”; “certified” is a wrong word here.

p.4, first line “mostly unstructured protein.”
Reference for that? I don’t recall any knowledge about that.

p.4, last but 6th line “… into a monomeric species by binding it to …”
Bad English: “binding it to”.

p.5, line8 “dimerization/aggregation”
Try to avoid use of stroke in formal English writing.

p.6, lines 3 and 4
Missing a few spaces characters here.

p.7, line 6 “deletion constructs for expression analysis”
Misleading here. “expression analysis” may mean an analysis of the level of expression of a construct which clearly is not here.

p.7, lines 12-14 “The YFP fusions of ataxin-1 …not influence the data.”
Similar expression pattern is not proof that a fusion protein which carries an extra moiety does not influence the data. It literally just mean that the two proteins are expressed at similar patterns. The fusion protein may have a totally disrupted structure.
p.7, last but 4 line, “The observation that … is able of …”
“is able of”: bad English.

p.7, last sentence
These results cannot rule out polyQ may be involved in foci formation. It can only be said that polyQ does not play a major role. It may be partly involved but not stable enough to sustain a detectable interaction with the rest (e.g. CT2AXH) deleted.

p.19, line 1, Fig. legend for Fig. 4 “… (panels C,D and E).”
Should be “… (panels D and E).”

p.20, Legend for Fig. 7
What does the asterix represent?

Figure 1
Needs to be revised. A number of places that should be aligned are out of vertical alignment.
-CT2AXH position 227
-Position 410 of TLND2End and of TLND2AXH
-The positions 562 and 689 of Atx1 and all the rest
-All the positions of 816
-Several constructs have horizontal alignment badly done too. The lines are not horizontal.

Symbols inconsistent: AXH2END vs AXH2End; TLND2END vs TLND2End (Table 1 and in the text).

Incorrect phases: “extra nuclear” should be extranuclear

---

## Round 0.2 · Minor Revisions

I looked through your manuscript to see why the reviewer still wants you to make few minor changes and I agree with the reviewer.
It looks like you made a mistake in labeling AXH and TLND2AXH in Fig. 5, at least it would explain the discrepancy between the figure and the text. But if there is no mistake then address the reviewer's concerns about your explanation of these results.

In the legend for the figure 7 you should write about statistical analysis you used.

·

Basic reporting

No comments.

Experimental design

No comments.

Validity of the findings

In rebuttal:
“Indeed, it was our original intention to do the experiment with the CT2AXH construct. However, we ran into problems of expression as the protein went into inclusion bodies. For this reason we switched to TLND2AXH, which expressed well and was soluble.”
Can the authors put a sentence in the results as a record?

Line 169 “Identified of a self-…”
Should be “Identification of a…”

SEC results:

I am repeating what I commented previously. Lines 176-178 “Analytical gel-filtration … the construct TLND2AXH is in equilibrium between two species which have an elution volume compatible with that of a monomer and a dimer.”
The 2 peaks of TLND2AXH are of 11.0 and 12.0 ml, these correspond to roughly 36 kDa and 30kDa respectively if it were a globular protein (from the standards). That is all one can say. It cannot be claimed to be compatible with that of a monomer and a dimer of a 16 kDa unstructured protein (without proper calibration).

In rebuttal :
“…We have now corrected the text to indicate lower and not higher migration.”
Are we talking about the same thing? I don’t see this corrected. In Lines 178-179: “Higher retention volumes are observed for TLND2AXH as compared to AXH.” I think the solid line (AXH) has higher retention volumes.

Lines 181-183: “The elution volume (12.7 ml) of the higher molecular weight species of TLND2AXH is anyway significantly smaller that expected for a trimer (48 kDa) or a tetramer (64 kDa).”
Again, I am repeating this comment. I can only see a 12.7 ml peak belonging to the continuous line of AXH domain (if it was labelled correctly), not of TLND2AXH (dotted line). To my eyes, the higher MW species of TLND2AXH has an elution volume of 11.0 ml (NOT 12.7ml!) and corresponds to ~36kDa globular protein. I suggest the author not to mention the trimer or tetramer as this is again going back into the issue of comparing MW of unstructured and globular species.

In rebuttal:
“Our point here was that the two constructs are of comparable size (ca. 15 kDa). It does therefore make sense to QUALITATIVELY ...”
I agree that the 2 species of similar MW can be qualitatively compared if they are similar. However, since the two species may have different oligomerisation states (AXH is globular and dimer/tetramer whereas TLND2AXH is unstructured and has two states but not necessarily monomer/dimer), I think it is not a good idea to qualitatively compare these. As it is, it is quite confusing. May I suggest the authors either do proper quantitative comparison or the SEC results can be left out without affecting the findings and conclusion of this work.

p. 20, Legend for Fig. 7. What does the asterix represent?
Rebuttal: “We apologies but we cannot see the asterix.”
The stars on top of each pair of data. Obviously it refers to the statistical significance of the difference, and ns means “not significant”. But all these need to be written out in the legend.

Additional comments

No comments.

---

## Round 0.3 · accepted · Accept

( there are no comments )